# Hypoxic Environment and Paired Hierarchical 3D and 2D Models of Pediatric *H3.3*-Mutated Gliomas Recreate the Patient Tumor Complexity

**DOI:** 10.3390/cancers11121875

**Published:** 2019-11-26

**Authors:** Anne-Florence Blandin, Aurélie Durand, Marie Litzler, Aurélien Tripp, Éric Guérin, Elisa Ruhland, Adeline Obrecht, Céline Keime, Quentin Fuchs, Damien Reita, Benoit Lhermitte, Andres Coca, Chris Jones, Isabelle Lelong Rebel, Pascal Villa, Izzie Jacques Namer, Monique Dontenwill, Dominique Guenot, Natacha Entz-Werle

**Affiliations:** 1Laboratory EA3430. Progression tumorale et microenvironnement, Approches Translationnelles et Epidémiologie, University of Strasbourg, 3 avenue Molière, 67000 Strasbourg, France; aurelie.durand@unistra.fr (A.D.); marie.litzler@gmail.com (M.L.); aurelien.tripp@gmail.com (A.T.); eric.guerin@chru-strasbourg.fr (E.G.); damien.reita@chru-strasbourg.fr (D.R.);; 2Department of Nuclear Medicine, University Hospital of Strasbourg, 1 avenue Molière, 67098 Strasbourg, France; eruhland@live.fr; 3PCBIS Plate-forme de chimie biologique intégrative de Strasbourg, UMS 3286, CNRS, Université de Strasbourg, Labex Medalis, 300 boulevard Sebastien Brant, F-67000 Strasbourg, France; aobrecht@unistra.fr (A.O.); pvilla@unistra.fr (P.V.); izziejacques.namer@chru-strasbourg.fr (I.J.N.); 4Institut de Génétique et de Biologie Moléculaire et Cellulaire, CNRS UMR 7104, Inserm U964, 1 rue Laurent Fries, 67400 Illkirch, France; keime@igbmc.fr; 5UMR CNRS 7021, Laboratory Bioimaging and Pathologies, Tumoral Signaling and Therapeutic Targets, Faculty of Pharmacy, 74 route du Rhin, 67401 Illkirch, France; Quentin.fuchs@unistra.fr (Q.F.); benoit.lhermitte@chru-strasbourg.fr (B.L.); isabelle.lelong-rebel@unistra.fr (I.L.R.); monique.dontenwill@unistra.fr (M.D.); 6Pathology Department, University Hospital of Strasbourg, 1 avenue Molière, 67098 Strasbourg, France; 7Centre de Ressources Biologiques, CRB, University Hospital of Strasbourg, 1 avenue Molière, 67098 Strasbourg, France; 8Neurosurgery, University Hospital of Strasbourg, 1 avenue Molière, 67098 Strasbourg, France; hugoandres.coca@chru-strasbourg.fr; 9The Institute of Cancer Research, 15 Cotswold Road, Sutton, Surrey SW7 3RP, UK; chris.jones@icr.ac.uk; 10Pediatric Onco-Hematology Department, Pediatrics, University hospital of Strasbourg, 1 avenue Molière, 67098 Strasbourg, France

**Keywords:** models, pediatric high-grade glioma, hypoxia, intra-tumor heterogeneity, tumor and cell metabolism

## Abstract

Background: Pediatric high-grade gliomas (pHGGs) are facing a very dismal prognosis and representative pre-clinical models are needed for new treatment strategies. Here, we examined the relevance of collecting functional, genomic, and metabolomics data to validate patient-derived models in a hypoxic microenvironment. Methods: From our biobank of pediatric brain tumor-derived models, we selected 11 pHGGs driven by the histone *H3.3K28M* mutation. We compared the features of four patient tumors to their paired cell lines and mouse xenografts using NGS (next generation sequencing), aCGH (array comparative genomic hybridization), RNA sequencing, WES (whole exome sequencing), immunocytochemistry, and HRMAS (high resolution magic angle spinning) spectroscopy. We developed a multicellular in vitro model of cell migration to mimic the brain hypoxic microenvironment. The live cell technology Incucyte^©^ was used to assess drug responsiveness in variable oxygen conditions. Results: The concurrent 2D and 3D cultures generated from the same tumor sample exhibited divergent but complementary features, recreating the patient intra-tumor complexity. Genomic and metabolomic data described the metabolic changes during pHGG progression and supported hypoxia as an important key to preserve the tumor metabolism in vitro and cell dissemination present in patients. The neurosphere features preserved tumor development and sensitivity to treatment. Conclusion: We proposed a novel multistep work for the development and validation of patient-derived models, considering the immature and differentiated content and the tumor microenvironment of pHGGs.

## 1. Background

Pediatric high-grade gliomas (pHGGs), including diffuse intrinsic pontine glioma (DIPG), are devastating diseases [1]. They are the leading cause of mortality in pediatric oncology. These tumors exhibit progression after current treatments, with more than 90% of children relapsing within two years of diagnosis [2,3]. Since 2010, the histone *H3.3* and *H3.1 K28M* mutations have been reported as recurrent driver events in midline pHGGs. The significance and function of these aberrations have not been clearly established, but are now diagnostic tools for pHGGs [4,5]. The identification of new effective therapies is challenging, currently relying on low or large-throughput screens in the laboratories and the lack of in vitro models entirely recapitulating the pHGG tumor diversity [6,7].

Therefore, patient-derived tumor cell line (PDCL) models in pHGGs are becoming the new standard for the preclinical drug testing and biomarkers’ discovery at diagnosis or on autopsy specimens [8,9]. In addition to traditional 2D monolayer (MNL) cell cultures, the 3D models, such as neurospheres (NS), multicellular tumor spheroids, and organoids, are valuable for studying brain tumors, as they contain several cell types and harbor a more complex 3D anatomy [9,10,11,12]. Recent publications have mainly investigated the efficacy of new targeted therapies in 3D PDCLs and/or in patient-derived xenografts (PDX) [8,13,14,15].

As presented in a recent paper [16], the pHGG bearing the *H3.3 K28M* mutation mainly contains an oligodendrocyte-like signature and more differentiated malignant cells, including differentiated astrocytic-like cells. Therefore, we believe that the isolation of those cell types (2D and 3D cultures) from the same tumor patient might accurately recreate the multilineage organization of the pHGGs to study the drug response in vitro. This will take into account the differentiation hierarchies of tumor cells involved in tumor development and in drug resistance [16,17,18,19,20].

Since 2010, our laboratory initiated the PEDIAMODECAN (PEDIAtric MODEls for CANcer research) program to develop and characterize novel patient-derived models from several pediatric brain tumor types. Here, we propose a detailed characterization of the 3D and paired 2D cellular models derived from midline pHGGs, driven by the histone *H3.3K28M* mutation. We looked at comprehensive genome profiling, as well as neural/glial components in 2D/3D PDCLs and patient tumors. Metabolomic characteristics were evaluated using HRMAS (high resolution magic angle spinning) analysis in our PDCLs models maintained in hypoxia (5% O_2_) and in a cohort of *H3.3-K28M*-mutated pHGGs. We developed a hypoxic (1% O_2_) spheroid model to mimic the cell dissemination from a tumor mass observed during patient tumor progression. We also performed intracranial xenografts of 3D and 2D PDCLs in immunocompromised mice. Finally, we exposed all cell lines to low and high level of oxygen, and treated them with the standard alkylating agent temozolomide, as no innovative drugs are currently efficient against those pHGGs.

This study sought to classify 2D/MNL PDCLs, as immature differentiated astrocytic cells, as being able to participate in slow tumor growth and cell migration, and the paired 3D/NS PDCLs cultures, as tumor initiating-like cells, as being able to restart tumors in case of minimal residual cells. The final goal was to propose a standardized method to develop pHGGs models for drug testing and recreate an accurate brain environment in vitro.

## 2. Results

### 2.1. Clinical Features of the H3.3-K28M-Mutated High-Grade Gliomas

Among patients included in the PEDIAMODECAN program (seven DIPGs and four tHGGs (thalamic high-grade gliomas)), we selected four cases that were mismatch repair proficient and harboring the driver histone mutation *H3.3 K28M*. We were able to derive four PDCLs out of seven DIPGs and two PDCLs out of four tHGGs. Among the four DIPG PDCLs, we had only two *H3.3-K28M*-mutated tumors. The patient ages ranged from 9 to 18 years old. The clinical reports, including immunohistochemistry results, are summarized in Table 1. MRI radiological data showed that DIPGs and tHGGs invaded the pons or thalamus, respectively, at diagnosis, with a localized tumor enhancement after gadolinium injection on T1-weighted and T2-weighted FLAIR (fluid-attenuated inversion recovery) sequences. The immunohistochemical analyses were performed for Ki67, GFAP (glial fibrillary acidic protein), p53, EGFR (epidermal growth factor receptor), PTEN (phosphatase and tensin homolog), Olig2 (oligodendrocyte transcription factor 2), and HIF-1α (hypoxia inducible factor 1 alpha) biomarkers.

### 2.2. Early-Passage Culture of PDCLs Preserved the Genotype and Phenotype of Primary High-Grade Gliomas

Upon receipt of the fresh tumor material of each patient, we generated concomitantly 3D/NS and 2D/MNL cell cultures from BT68, BT69, and BT83 tumors. BT35 cells grew only in a 2D/MNL line. To assess whether the PDCLs features matched to the patient tumor, we first performed comprehensive genome profiling. The targeted-sequencing analyses showed the maintenance of the somatic driver *H3.3 K28M* in all MNL and NS PDCLs for all passages (Figure 1a,b). The VAF (variant allele frequency) of the heterozygous histone mutation *H3.3 K28M* increased in cell lines compared to the paired patient tumor, suggesting the in vitro expansion of PDCL-selected mutated tumor cell populations. As genetic drifts in culture have long been documented, we performed whole-exome sequencing for early passages (<10) in the cell lines and compared them to the patient tumors. The numbers of DNA alterations increased in both 2D and 3D lines compared to the tumor sample after multiple passages (Figure 1c, Appendix A). Nevertheless, PDCLs preserved the main genomic aberrations found in patient tumors, including histone, *ATRX* (alpha-thalassemia/mental retardation syndrome X-linked) mutations, and *CDKN2A* (cyclin-dependent kinase inhibitor 2A) homozygous deletion. The expression of the tumor glial marker GFAP and the neural marker Nestin were maintained in MNL and NS cultures for early passages, while long-term cell expansion partially lost the markers’ expression (Figure 1d). In the data in Appendix A, we report the persistent expression of stem-cell marker SOX2 (SRY-Box2), CD133, and OCT4 (octamer-binding transcription factor 4), the neuron-specific progenitor marker MAP2 (microtubule associated protein 2), and the oligodendroglial markers O4 or Olig2 in NS PDCLs, characterizing the immature neuronal potential of the NS cells with an oligodendroglial-like pattern. Olig2 decreased in the tumor relapses for BT35 and BT69 (Table 1), as in the corresponding cell lines. The MNL lines mostly preserved more mature astrocytic markers, like GFAP and the mesenchymal marker CD105.

We next considered how the growth condition related to genetic heterogeneity. As expected, CNV (copy number variation) analyses did not reveal unique genetic imbalance signatures for 2D/MNL or 3D/NS in the examples BT69 and BT83 (Appendix A). We noticed that the MNL culture mostly led to a gain in chromosome alleles (extra-chromosomal abnormalities 1, 7, 9, 11, and 12 in BT69 MNL and extra material of chromosome 16 in BT83 MNL) (Appendix A). The NS culture harbored predominantly chromosomal losses (loss of material on chromosomes 10, 15, and 18 in BT69 NS and interstitial loss on chromosome 1 in BT83 NS). Overall, BT35 MNL harbored gains of genetic material compared to the original tumor, while BT69 NS presented similar CNVs to the patient tumor (Appendix A).

All those results confirmed our choice to work with both MNL and NS PDCLs for early passages (less than 10). They were appropriately matching to the genomic properties and protein markers of the patient tumor.

### 2.3. Importance of In Vitro Hypoxia to Mimic the Metabolic Changes During H3K28M-Mutated pHGG Progression

To go further and choose the more accurate culture environment, we explored the metabolomic profiles of our cell lines and tumor collections using HRMAS NMR spectroscopy. The tumor collection included six tHGGs and four DIPGs. All six tHGGs matched with their relapse samples. Two tHGGs were previously included to the PEDIAMODECAN program but we did not have a paired PDCL. The relapse specimens were obtained systematically at first recurrence following a Stupp protocol with 12 courses of post-radiotherapy. Only four tumors out of the seven DIPGs’ PEDIAMODECAN program harbored the driver mutation and were included in the in vivo HRMAS assessment.

Using ADEMA (Algorithm to Determine Expected Metabolite Level Alterations Using Mutual Information) analyses, we compared each cell line (passage 5) after 48 hours normoxic (21% O_2_) or hypoxic (5% O_2_) exposure, as well as paired tumors at diagnosis and relapse (Figure 2a,b and Appendix A). We showed that hypoxic conditions were significantly closer to the metabolism observed at relapse in the entire cohort of *H3.3*-mutated pHGGs (Figure 2a). The metabolic profile of BT69 NS at 5% O_2_ matched with the metabolism of BT69 at relapse but was quite different from the diagnostic sample (Appendix A). Indeed, PCho (14), NAA (13), glutamate (7), and glutamine (8) peaks were no longer present in the relapse sample and in the BT69 line. A low concentration of serine was also preserved in the BT69 line in hypoxia. All those observations in BT69, but also in BT83 and BT68 (Figure 2a and Appendix A), characterized a metabolic switch through lipolysis, serinolysis, and glutaminolysis, were also described in relapsing *H3.3*-mutated pHGGs (Figure 2b). RNA sequencing data confirmed the presence of these three metabolic pathways in PDCLs with a high expression of glycolytic enzymes in all four PDCLs, whereas glutaminolysis and serinolysis were particularly overexpressed in NS BT68, BT83, and BT69 (Appendix A). The Inginuity Pathway Analysis (IPA^©^) software analyses of RNA sequencing revealed that the glutamine metabolic pathway was predominant in NS cells comparatively to MNL, with an increased expression of glutamate ionotropic receptor AMPA (α-amino-3-hydroxy-5-methyl-4-isoxazolepropionic acid), glutamate receptor interacting protein 1, and glutamate metabotropic receptor 8. The amount of lactate also decreased in hypoxic 3D PDCLs and in tumor relapses, showing a less glycolytic switch during the pHGG progression associated to an increase of ROS (reactive oxygen species) regulation (Figure 2b), and a decrease of glycine and myoinositol and the glycine/myoinositol ratio (Appendix A). This ROS metabolism increase was significantly associated with histone *H3.3* modulation in 2D cells with IPA^©^ analyses. The FDG (fluorodesoxyglucose) PET-scans (positron emission tomography-scans) of BT69, BT68, and BT83 patients over the progression confirmed the significant decrease of the FDG uptake related to the decrease of glycolysis, as seen in the PDCL models (Figure 2c).

The metabolic profile of BT35 MNL in the hypoxia condition also matched the BT35 tumor metabolic profile at relapse (Appendix A) and was significantly different from the diagnostic tumor. Nevertheless, this PDCL model was significantly different from the others (ADEMA analyses in Figure 2a), as its metabolism was more in favor of lipolysis and a less active amino acid metabolism (Appendix A).

Global metabolomic profiles showed that the two cell lines BT35 and BT69, derived from tHGGs, (R2Y = 0.95; Q^2^ = 0.85), and the two cell lines BT68 and BT83, derived from DIPGs (R2Y = 0.99; Q^2^ = 0.90), were clearly separated by a two-component PLS-DA (Partial Least Squares Discriminant Analysis) analysis of the global metabolite profiles (Appendix A). Nevertheless, the specific metabolism changes in cell lines with 5% O_2_ was similar to the paired tumor (Figure 2b) and the PET-scans (Figure 2c), confirming the major impact of hypoxia (5% O_2_) on cell cultures.

### 2.4. In Vitro Modeling of Cell Migration from Hypoxic PDCL Spheroids’ Recapitulated Tumor Cell Dissemination in Patients

To mimic the different tumor areas (regions that were well-oxygenated close to the perivascular space and the hypoxic regions distal to this neovasculature), we investigated PDCL cellular changes occurring in response to a strong decrease in oxygen level (from 21% to 5% and 1% O_2_). Following the deprivation of oxygen, we showed a progressive cellular detachment of the semi-adherent BT68 NS and the HIF-1α expression increase (Figure 3a). In BT68, the expression of HIF-1α was transiently upregulated after 48 h of low oxygen exposure compared to the UW479 control line (Appendix A). All cell lines highly expressed HIF-1α at 1% O_2_ at 48 h (Appendix A), but surprisingly, the cell phenotypes of BT35 MNL, BT69 NS, and BT83 NS were not modulated under hypoxia.

Transcriptomic analyses identified a seven-gene hypoxia signature in PDCLs exposed to 1% O_2_ (*p* < 0.01 and log2 fold change > 2) (Figure 3b). The hierarchical clustering revealed only an increased expression of these genes in NS cultures, suggesting that the 3D/NS culture better reflected the pathological hypoxic gradient of brain tumors. CA9 (carbonic anhydrase IX) and VEGFA (vascular endothelial growth factor A) are well-known markers of hypoxia. For the other biomarkers, AK4 (adenylate kinase 4), PDK1 (pyruvate deshydrogenase kinase 1), and HK2 (hexokinase 2) are metabolic enzymes involved in glycolytic pathways. mir210HG and BNIP3 (BCL2/adenovirus E1B 19-kD interacting protein 3) are usually expressed in hypoxic conditions [21].

To understand the invasiveness of PDCLs in the hypoxic brain environment, we developed a spheroid in vitro model (Figure 3c) recapitulating hypoxia observed in patient brain tumors. Because the oxygen diffusion ranged from 70 µm to 200 µm in tumors and normal tissue, respectively [21], we generated spheroids (from early passages (<10) of paired MNL and NS lines) with diameters above 200 µm to create a hypoxic gradient from the periphery to the core and expanded them at 5% O_2_. We showed that the expression of the stem cell marker SOX2 was restricted to the periphery of the spheroid, while HIF-1α was expressed from the core to the periphery of the spheroid. We confirmed the co-expression of HIF-2α/HIF-1α and SOX2 observed in HGGs (Figure 3c). Furthermore, previous studies, comparing gene expressions in HGGs versus normal brain parenchyma, identified the extracellular matrix protein fibronectin as one of the most over-expressed gene in HGGs [22]. Then, to more carefully recreate the biology of the patient’s tumors, we immobilized the spheroids onto a fibronectin coating. The quantification of cells evading from the spheroid and the distance of migration showed that cell dissemination was slower in MNL than in their corresponding NS lines (Figure 3d). In addition, the invasiveness of each patient tumor during its progression or relapse was comparable to the migration profiles in MNL- and NS-derived spheroids (Figure 3e). As shown on MRI, the highly diffusive BT69 tumor relapse in the left thalamic initial site and in the contralateral anterior ventricle was similar to the highly invasive profile of the BT69 spheroid (Figure 3e,f). The localized BT83 tumor at relapse was comparable to the low diffusive BT83 spheroid (Figure 3e). Similar comparisons could be performed for BT68 and BT35. The SUV (standardized uptake value) on PET scans decreased in proportion to the increase of cell migration potential in the spheroid models (Figure 2c). This approach seems to confirm the importance of oxygen level variations to study in vitro cell behavior and migration.

### 2.5. Both 2D/MNL and 3D/NS Cell Cultures Mimic the Hierarchical pHGG Development and Patient Drug Sensitivity

To go further in the comparison of NS and MNL features and to understand the functional impact of hypoxia, we performed a proliferation assay using the IncuCyte^©^ system. In all PDCLs, the MNL cells harbored a significantly longer doubling time than the corresponding NS populations in both normoxic and hypoxic environments (Figure 4a). The proliferation was slightly higher in hypoxia with an average doubling time ranging between 3 and 18 days in normoxia and between 1.6 and 17 days in hypoxia. Surprisingly, the doubling time of BT35 adherent cells was similar to cells growing as NSs. We then investigated the PDCLs clonogenic potential (Figure 4b) and showed that only NS cultures formed colonies (plating efficiency: 11%, 17%, and 12% in BT69, BT68, and BT83, respectively). By contrast, 2D/MNL cultures failed to form colonies, except for BT35 cells, which developed diffuse colonies (plating efficiency: 11%).

In order to understand why 2D/MNL PDCL cannot form colonies, we compared the gene expression profiles of MNL and the paired NS using RNA sequencing. We identified 452 genes that were differentially expressed. A total of 200 genes were upregulated (FDR (false discovery rate) > 0.05) and 252 were downregulated (FDR < 0.05) more than 3-fold (Figure 4c). Functional annotations of the upregulated gene set using GSEA (gene set enrichment analysis) identified significantly enriched biological processes in NSs (involved in central nervous system development, neurogenesis, and cell–cell adhesion) (*q*-values ranged from 8.7 × 10^−11^ to 1.7 × 10^−14^). The most downregulated genes in NS cells were implicated in extracellular matrix organization, tissue development, and differentiation (q-values ranged from 3.9 × 10^−19^ to 8.9 × 10^−41^). In addition, we isolated nine significant upregulated genes that are well-known to be involved in stem-like cell potential only in 3D/NS cultures (Figure 4d).

We then explored the tumor initiation properties of each cell line in NSG (NOD scid gamma) mice. We showed that NS PDCLs led to tumor development by an average of 80 days following the intracranial injection. We did not observe tumor formation from any MNL PDCLs (Figure 4e). The tumor mass histology recapitulated the hallmarks of the tumor patient. GFAP-positive cells were invading the mouse brain parenchyma at the periphery of the tumor mass (Appendix A). Targeted sequencing confirmed the presence of the *H3.3 K28M* histone mutation in all NS xenografts (Appendix A). Histological analyses might be a limited methodology to detect early tumor formation or disseminated single cells [23]. To improve the detection of PDCL infiltrating cells within mouse brain, we quantified the human-specific *Alu* repeats expressed in mouse brain tissue at the injection site **(**Appendix A). We quantified 0.2%, 22.2%, and 100% of *Alu* sequences in BT69, BT83, and BT68 NS, respectively. We confirmed that mice injected with MNL cells did not exhibit any tumor development, even BT35, harboring clonogenic properties and a high proliferation rate.

We then compared the sensitivity of the 2D and 3D models to temozolomide (TMZ), a chemotherapeutic agent targeting cycling cells and used in the past for pHGG treatments (Figure 4f and Appendix A). At 21% O_2_, all NS PDCLs were sensitive to TMZ after long-term exposure (120 hours for BT69, 152 hours for BT83, and 135 hours for BT68), whereas the paired-MNL lines were completely resistant. We also showed that a large proportion of NSs expressed the mitotic marker phospho-H3ser10, while MNL poorly expressed it, which is consistent with the higher proliferative rate observed in NSs. This might partially explain the absence of MNL response to TMZ. As expected, PDCLs derived from HGG were resistant to TMZ, but we showed that, surprisingly, hypoxia increased the 3D/NS response to high doses of TMZ (Appendix A). Indeed, the IC50 of BT83 and BT69 were lower at 5% O_2_ than 21% O_2_ (BT83 IC50^(5%O2)^ = 74 µM at 96 h; BT83 IC50^(21%O2)^ > 200 µM at 96 h; BT69 IC50^(5%O2)^ = 148 µM at 48 h; BT69 IC50^(21%O2)^ = 200 µM at 48 h) (Appendix A). The BT35 cell line was completely resistant in all conditions. Those observations in hypoxia were confirmed in clinic as the patient BT69 responded to TMZ for 12 months. Nevertheless, it also confirmed that it is possible a part of pHGG tumors depicted by MNL PDCLs were completely resistant from diagnostic to this alkylating agent and might be tested systematically.

## 3. Discussion

Research on pHGG models are now a priority as there is clearly a lack of in vitro models entirely recapitulating the pHGG tumor diversity. For this purpose, here we are proposing a standardization of patient-derived cell lines to integrate tumor characteristics, as well as part of the brain microenvironment (Figure 5a).

The first step was to establish the optimal culture conditions. We expanded the 2D cells by plating them on a laminin coating substrate in DMEM (Dulbecco’s Modified Eagle’s Medium)/F12 GlutaMAX medium supplemented with serum. The increase of serum from 1% to 10% did not change the 2D characteristics, but that helped to expand the 2D cultures above 15 passages. We tested the laminin coating alone without serum and the cells stopped growing. We expanded the 3D culture using a standard combination of DMEM/F12 GlutaMAX medium supplemented with B27, human Fibroblast Growth Factor-basic (bFGF) and recombinant human epidermal growth factor (EGFR).

In those conditions, both the 2D and 3D PDCLs early passages preserved the genomic drivers and neural/glial components observed in pHGGs, while long-term cell cultures often lacked these features. Both MNL and NS models might be used for drug screening purposes. They are capable of representing all tumor cell subclones, combining self-renewal and tumor growth properties (Figure 5b) [24]. To go further, we developed a concomitant 2D/3D culture that preserved the differentiated and stem-like subpopulations after 10 passages (Figure 5c). We will next generate xenograft models with this co-culture, as suggested by Wang et al. [19] for adult HGGs [16].

Our metabolic analyses and spheroid modeling also confirmed that a hypoxia environment closely reflected the metabolic reprogramming of a patient tumor and their progression, and might be a key component of in vitro models to more closely resemble the state of the patient (Figure 5a). To our knowledge, HGG-derived cell lines are always expanded in a controlled well-oxygenated environment (21% O_2_), which does not reflect the oxygen level of the brain microenvironment [25]. Indeed, pHGGs are highly hypoxic tumors (1% O_2_) and the oxygen pressure of the normal brain is above 35 mmHg, corresponding to 5% oxygen. Therefore, drug screening performed at 21% O_2_ probably underestimates the microenvironment impact. Here we described that BT lines grew faster at 5% O_2_ than under the usual normoxic atmosphere. Cell growth determines the response to cell-cycling-targeted therapy. Therefore, expanding pHGG-derived cell lines at 5% O_2_ might mimic the cell growth in the patient more closely (Figure 5a). The hypoxia also induced an altered metabolism, which is considered a feature of adult glioblastoma progression, but the mechanisms by which it contributes to oncogenicity are still unknown, especially in pediatric patients. We [26] and others [27,28,29,30] described this HGG metabolic heterogeneity and how it might impact the response to therapeutic drugs. Our data also explored the process of compensatory metabolism in pHGG tumors and the paired-cellular models. We described a metabolic switch between the tumor at diagnosis and the matched tumor at relapse (decrease of glycolysis and increase of lipolysis, serinolysis, and glutaminolysis). We demonstrated that growing cells at 1% O_2_ more closely reflected this patient tumor metabolism than cells expanded in normoxy.

In addition, for drug testing, most of the studies in the literature focused on 3D/NS PDCLs, which were considered the gold standard cultures to preserve stem-like features [31] and resistance to therapy. Some other publications supported an equivalence of 2D and 3D cultures for modeling drug response [17,32]. Others articles described the genomic differences of paired adult HGG NSs and MNLs, as observed in our experiments, and how the collaboration between subclonal cell populations determines drug response [18,19,33,34]. We demonstrated here that 3D/NS were sensitive to a high dose of TMZ, while the paired-MNL cells were totally resistant to this drug. Those results might be explained by our functional and genomic data identifying 2D/MNL as immature differentiated cells with a low in vitro expansion and 3D/NS as a progenitor-like subset of cells with a high proliferation rate. The argument of previous studies to prioritize 3D models was also in terms of the ability to restart tumors in animal models. We showed also that only 3D/NS models led to tumor development in mice brains, but this limited in vivo model will not be able to entirely describe the pHGG cellular complexity [35,36].

To understand how the oxygen level might interfere with drug testing, we chose a standard drug used in the past, as no innovative drugs, for instance, are efficient against those pHGGs. We assessed the 2D and 3D paired lines response to the alkylating agent TMZ in normoxy and hypoxia. Surprisingly, a complete resistance of 2D PDCL was observed in all cell lines in both oxygen environments. By contrast, 3D cells were differentially sensitive in normoxic and hypoxic environments, highlighting a low level of oxygen as a key in the modeling of pHGGs resistance/sensitivity.

We already knew that the real benefit of TMZ treatment was limited by acquired resistance, especially in DIPGs and more recently in thalamic pHGGs [2,37]. In the last decade, many publications have explored resistance mechanisms to TMZ, including the tumor heterogeneity, MGMT (O-6-methylguanine-DNA methyltransferase), and mismatch repair status, but these processes have not completely explained this lack of sensitivity in pediatric patients. Therefore, the 2D/3D cell model and the oxygen modulation might help us to understand the drug resistance mechanisms and pave the way for new therapies in pHGGs.

In conclusion, our in vitro and in vivo data classified 2D/MNL PDCLs, as immature differentiated astrocytic cells, as being able to participate in the slow tumor growth and cell migration, and paired 3D/NS PDCLs cultures, as tumor initiating-like cells, as being able to restart tumors in the case of there being minimal residual cells. Combining the data of 2D and 3D PDCLs of the same patient, a pHGG might preserve the tumor hierarchy and complexity. It will also guide new aspects in preclinical drug screening concomitantly with the hypoxic microenvironment.

## 4. Methods

### 4.1. Patient Tumors and Derived-Cell Cultures

All samples were obtained after the informed consent of parents and patients was given. Specimens were anonymized for their analyses. This study was conducted in accordance with the ethical committee approval. Patients’ characteristics of BT35, BT68, BT69, and BT83 are in Table 1. Biopsies were cut into 1 mm^3^ pieces and dissociated using a GentleMACS dissociator (Miltenyi Biotec, Paris, France). Cell suspensions were treated with Red Blood Cell Lysis solution (130-094-183, Miltenyi Biotec). For the NS/3D culture, cells were plated in DMEM/F12 GlutaMAX medium supplemented with B27 (Gibco, Fisher Scientific, Illkirch, France), human Fibroblast Growth Factor-basic (bFGF, Millipore, Fontenay sous Bois, France), and recombinant human Epidermal Growth Factor (HuEGF, Gibco). MNL culture was established by plating glioma cells on a laminin coating substrate in DMEM/F12 GlutaMAX medium supplemented with 10% serum. Combined 2D/3D cultures were based on DMEM/F12 GlutaMAX with HuEGF and bFGF, as well as FBS (fetal bovine serum) 4% and B27 2%. The hypoxic conditions were set up in an incubator Sanyo MCO 18M.

### 4.2. Next Generation Sequencing (NGS) Analyses

DNA extractions from patients’ tumors, cultured cells, and stereotaxic xenograft frozen tissues (50 µm × 5) were quantified using a fluorimetric method (Qubit dsDNA BR Assay, Thermo Fisher Scientific, Illkirch, France) and qualified using real-time PCR (FFPE QC Kit, Illumina, San Diego, CA, USA). Mutation screening was performed using next generation sequencing on a MiSeq Illumina platform, using a Tumor Hotspot MASTR Plus assay (Multiplicom-Agilent, Santa Clara, CA, USA). A total of 26 gene mutations (hotspots) were screened, including H3F3A. Sequencing data were aligned to human hg19 using BWA-MEME algorithm (Burrows–Wheeler aligner-maximal exact matches). The minimum coverage per base (DP depth) and the minimum variant allelic frequency (VAF) were fixed at 500-fold and 5%, respectively. Data were visualized using Integrative Genomics Viewer (Broad Institute, USA). For BT35, BT68, BT69, and BT83, we also performed exome sequencing analyses using the SureSelect Human All Exon capture sets V4 (Agilent, Santa Clara, CA, USA), and paired-end sequenced on an Illumina HiSeq2000 with a 100 bp read length (median coverage = 100×).

### 4.3. Immunocytochemistry

A density of 2 × 10^4^ 2D cells/well were seeded onto glass coverslips (Thermo Fisher Scientific). Floating 3D/NS were embedded in optimal cutting temperature (OCT) mounting media (Sakura Tissue-Tek, Alphen aan den Rijn, The Netherlands). Cryosections (7 µm) were cut using a Leica CM3050S microtome (Leica Biosystems, Nanterre, France). Sections were stained with anti-Nestin Mouse (1:200, MAB5326 Millipore), anti-Nestin human (1:500, ABD69 Millipore), anti-GFAP (1:500 837,201, BioLegend, London, UK), anti-βIII tubulin (1:200, ab18207 Abcam, Paris, France), anti-O4 (1:100, 07, 139 Sigma, St. Quentin Fallavier, France), anti-MAP2 (1:500, AB5622 Millipore), anti-SOX2 (1:200, MAB2018, R&D System, Abingdom, UK), anti-phospho-histone H3 Ser10 (1:200, 06, 570 Millipore), together with an anti-mouse or anti-rabbit Alexa Fluor 488 and anti-rabbit Cy3 secondary antibodies (Abcam), using routine protocols. 

### 4.4. Immunohistochemistry (IHC)

IHC was performed using an automated tissue staining system (Ventana Medical Systems, Inc., Tucson, AZ, USA) on 4-µm-thick paraffin-embedded sections with an OptiView DAB IHC Detection Kit (detecting mouse IgG, mouse IgM and rabbit primary antibodies, Roche Diagnostics, Meylan, France). The primary antibodies were Ki67 (clone MIB-1, Dako, Santa Clara, CA, USA), GFAP (clone 6F2, Dako), p53 (clone BP53-12, Zytomed Systems, Berlin Germany), PTEN (138G6, Cell Signaling Technology, Ozyme, St-Quentin-en-Yvelines, France), EGFR (clone E30, Dako), HIF-1α (Abcam), and Olig2 (Abcam).

### 4.5. Array Comparative Genomic Hybridization (aCGH) Analyses

aCGH was performed using Agilent Human Genome G3 SurePrint CGH microarrays 4 × 180K (Agilent Technologies), following the manufacturer’s instructions. After all DNA extractions, 0.5 μg of sample DNA, along with reference DNA of the similar sex (Promega, Charbonnieres-les-Bains, France), was fragmented and labeled with a fluorescent tag, using the SureTag Complete Labeling Kit (Agilent Technologies, Santa Clara, CA, USA). The array slides were scanned at a 3 μm resolution on a Agilent Microarray Scanner System (Agilent Technologies). Agilent’s Feature Extraction software (V.1.5.1.0) extracted the data, which were analyzed with Agilent CytoGenomics software (v3.0.5.1) (Agilent Technologies).

### 4.6. Sample Preparations for NMR (Nuclear Magnetic Resonance) Measurements

Two-day-old cell cultures were incubated in a hypoxic (1% O_2_) or normoxic atmospheres (21% O_2_) for 48 hours. After dissociation, the cell suspension was washed with PBS to remove metabolites from the medium culture. Cell pellets of 1 × 10^6^ cells or fresh tumor samples (20 ng with ≥50% tumor cells) were collected into a 25 µL insert (B4494, Cortecnet, Brooklyn, NY, USA), frozen on dry ice, and stored at −80 °C until high-resolution magic angle spinning (HRMAS) analyses were performed. The tumor collection comprised 10 *H3.3 K28M* tumors at diagnosis and their 6 paired relapses. To provide a lock frequency for the NMR spectrometer, 10 µL of D_2_O was added to the insert. HRMAS NMR spectra were analyzed on a 500 MHz Bruker Avance III spectrometer (Bruker BioSpin, Billerica, MA). Data acquisition and spectra processing were performed as previously described [38]. A 1D proton spectrum was acquired for each sample. The chemical shift was referenced to the peak of the methyl proton of L-lactate at 1.33 ppm. The quantification procedure was based on the pulse-length-based concentration measurement (PULCON). Spectra were normalized according to each sample weight and calibrated using the signal intensity of a 3 µmol reference solution of lactate. Quantification results were expressed as nmol/mg of tissue.

### 4.7. RNA Sequencing Analyses

MNL and NS cell cultures were exposed to hypoxia, 1% O_2,_ or maintained in normoxic atmosphere (21% O_2_) for 48 hours. Total RNAs were extracted from cell lines using a Trizol reagent (ThermoFisher Scientific, Waltham, USA). Libraries suitable for high throughput sequencing were generated from 500 ng of total RNA using TruSeq Stranded mRNA Sample Preparation Kit (Illumina, Part Numbers RS-122-2101/RS-122-2102). DNA libraries were checked for quality and quantified using a Fragment Analyzer automated CE system (Advanced Analytical Technologies, Inc., Bath, UK). The libraries were sequenced on the Illumina HiSeq4000 system as single-end 1 × 50-base reads following Illumina’s instructions. Image analyses and base calling were performed using RTA 2.7.3 and bcl2fastq 2.17.1.14 (Illumina, San Diego, CA, USA). Reads were mapped onto the hg38 assembly of the human genome using TopHat2 pipeline [39] with the bowtie v2.1.0 (written in C++) aligner [40]. Gene expression was quantified using HTSeq v0.6.1 [41] and gene annotations from Ensembl release 87 [42]. Read counts were normalized across libraries with the method proposed by Anders et al. [43]. The data were published on the Gene Expression Omnibus website (www.ncbi.nlm.nih.gov/geo/) with the following accession number: GSE101799.

### 4.8. Western Blot Analysis

Two-day-old MNL and NS cell cultures were maintained at 21% O_2_ or at 1% O_2_ for 48 hours. Cells were lysed and subjected to an SDS-PAGE gradient. Immunoblots were probed with primary antibodies against HIF-1α (1:1000, ab51608 Abcam) and Actine (1:15000, MAB1501 Millipore) and anti-mouse or anti-rabbit (1:5000 NXA931, NA934V, GE Healthcare, Diegum, Belgium) secondary antibodies. Immunoblots were quantified with GeneTools analysis software (Syngene, Cambridge, UK).

### 4.9. Spheroid Migration Assay

Two-day-old spheroids (6000 cells/spheroid) were seeded onto fibronectin coating as previously described [10] and were transferred to the incubator with 5% O_2_. Twenty hours later, cells were fixed and stained with DAPI (4′,6-diamidino-2-phenylindole). Images of each well were acquired using a Zeiss macroscope (Axio Zoom V16, Munich, Germany). The number of evading cells was quantified with a home-made macro using ImageJ software (Java-based image processing program, NIH).

### 4.10. Doubling Time

MNL and NS cell lines were dissociated and plated in a regular 96-well plate or a U-bottom 96-well plate (Greiner Cellstar U-bottom culture plate), respectively (100 µL, 2000 cells). Fresh medium was added every week. Cell growth was followed for 15 days using IncuCyte^©^ Live Cell technology (Essen BioScience, Hertfordshire, UK) Pictures were automatically acquired over time and quantified using ImageJ software. Cell population doubling time was determined using the following equation: doubling time (days) = ln2(t−t_0_)/ln(N_t_/N_0_), where t − t_0_ is the time for exponential growth, and N_t_ and N_0_ are cell numbers at time t and t_0_, respectively, for 2D lines. For 3D lines, N_t_ and N_0_ are the neurosphere surfaces quantified by ImageJ software at time t and t_0_.

### 4.11. Clonogenic Assay

Cells were seeded onto six-well ultra-low attachment plates (500 cells/well) for 20 days. Floating formed colonies were immobilized onto a poly-L-lysine coating (19321-B, EMS, Hatfield, PA, USA) for 30 min at 37 °C and fixed in paraformaldehyde. Images of each well were acquired using a Zeiss macroscope (Axio Zoom V16). Colonies were manually counted using ImageJ software.

### 4.12. Intracranial Injections

Animal experiments were conducted in accordance with French guidelines for animal care and have been approved by the animal research committee (APAFIS #2017021410378167). A total of 300,000 cells resuspended in 2 µL PBS were injected into the left striatum of NSG (NOD Scid Gamma) mice. Animals were sacrificed at the onset of symptoms. Brains were excised to routine histological analysis.

### 4.13. Quantitative PCR (qPCR) of Human Alu Repeats

Genomic DNA was extracted from OCT sections localized at the injection sites (50 µm × 5). We used primers, which were specific for the Alu-J subfamily (sequences: F: 5’-CACCTGTAATCCCAGCACTTT-3’, R: 5’-CCCAGGCTGGAGTGCAGT-3’). qPCR was performed using the QuantiTect SYBR Green PCR Kit (Qiagen, Courtaboeuf, France) and the LightCycler system (Roche Life Science). Quantification of human DNA in murine tissue was based on a standard curve established from PDCL DNA. The linearity of standard curve ranged from 1 ng to 0.5 pg of human DNA. A detection of human DNA <0.5 pg was extrapolated and associated to background signal from human contaminations.

### 4.14. Drug Testing

Cells were plated in 96-well flat bottom plates (5000 cells/well). Cell confluency (% of occupied area) was imaged with an IncuCyte™ ZOOM cell imaging system (Essen BioScience) for 11 days and analyzed using Incucyte Zoom software. Temozolomide was used over a concentration range of 12.5 µM to 200 μM in comparison with controls and testing was done in triplicate for all cell lines.

### 4.15. Statistical Analyses

GraphPad Prism 6 software (GraphPad software, Inc.) was used for bar graph generation showing the mean +/− s.e.m. Comparisons between cell lines were performed using the non-parametric Kruskall–Wallis test followed by Dunn’s multiple comparison test. The paired-sample (MNL versus NS) comparisons were performed using a two-tailed *t*-test. A *p*-value < 0.05 was considered statistically significant and n.s. was reported for non-statistically significant results (*p* > 0.05).

Statistical comparisons from RNA sequencing data were performed using the method proposed by Love et al. [44] implemented in the DESeq2 Bioconductor library (v1.6.3) (Bioconductor). Adjustments for multiple testing was performed with the Benjamini and Hochberg method [45]. Functional annotation of the upregulated and downregulated gene sets was performed by Gene Set Enrichment Analysis (http://software.broadinstitute.org/gsea/index.jsp), as described in Subramamian et al. [46]. Complementary analyses were done using IPA^©^ (Ingenuity Pathway Analysis) software (Qiagen).

For metabolomics analyses, a multivariate network-based method, the algorithm to determine expected metabolite level alterations using mutual information (ADEMA), was used to statistically assess the changes in metabolite concentrations for all samples. We determined the relevant metabolites and grouped them using the metabolic network topology provided in the Kyoto Encyclopedia of Genes and Genomes [47,48] and the metabolic atlas by Selway et al [49]. The data set of the cell analyses was exported and analyzed into SIMCA P (version 13.0.3, Umetrics AB, Umea, Sweden) for a PLS-DA analysis. Two measurements were reported for the PLS-DA model: R^2^Y > 0.7 and Q^2^ > 0.5 for predicting the quality of the spectra and repeatability, respectively.

## 5. Conclusions

Collectively, our results highlight the importance of developing the early-passage of concomitant MNL and NS culture under the deprivation of oxygen to preserve divergent but complementary genomic drivers and neural/glial markers observed in pHGGs. This study provides evidence that hypoxic environment contributes to the maintenance of the original patient tumor metabolism and features. This is the first study combining protein expression, genomic, metabolism, and functional data as complementary criteria for characterizing patient-derived models. This multi-step work can be considered as a standard to develop clinically relevant model that faithfully recapitulates the human disease in pHGGs.

## Figures and Tables

**Figure 1 cancers-11-01875-f001:**
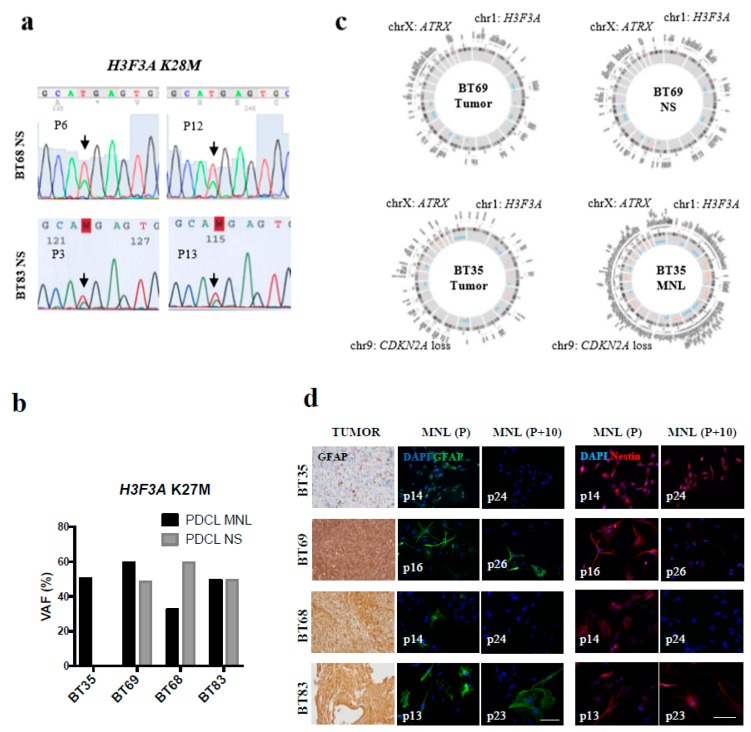
Patient-derived tumor cell lines (PDCLs) preserved genomic drivers and neural/glial markers’ heterogeneity observed in pediatric HGG, while long-term cell culture often lacked these markers. (**a**) Sanger sequencing at different culture passages (BT68 and BT83), illustrating the *H3F3A* mutation maintenance (arrows). (**b**) Allele frequencies of the heterozygous *H3F3A* mutation (Chr1: 226252135A > T; NM_002107: exon2: c.83A > T; p.K 28M) in the four PDCLs cultured as monolayer (MNL) and neurosphere (NS) (passages < 10). The variant allele frequency (VAF) was confirmed using two variant callers VarScan and gatkUG. (**c**) Circos plots of genetic alterations in patient tumors and the paired PDCLs. Point mutations, chromosome labels, cytoband information, and copy number variations (CNV) are shown from outer to inner circles. Deleted regions are depicted in blue and amplified regions are shown in orange. (**d**) Left panel: GFAP expression in thalamic high-grade gliomas (tHGGs; BT35 and BT69) and diffuse intrinsic pontine gliomas (DIPGs; BT68 and BT83) and the corresponding adherent cell lines at early (P) and late passages (P+10). Right panel: Immunodetection of Nestin. Scalebar—100 µm.

**Figure 2 cancers-11-01875-f002:**
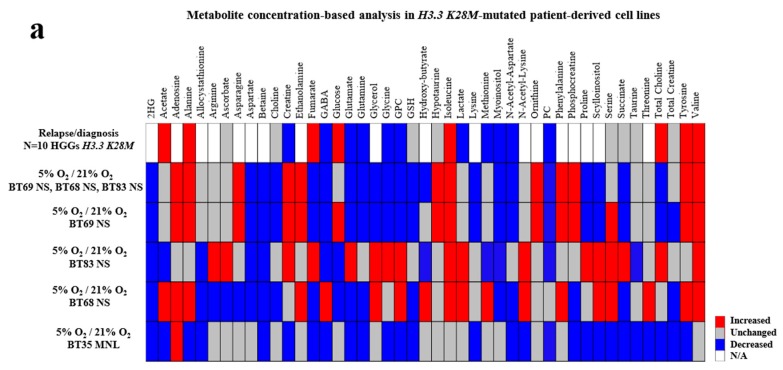
Ex vivo high-resolution magic angle spinning (HRMAS) of pediatric high-grade gliomas (pHGGs) and the impact of oxygen level on the paired-PDCL metabolisms. (**a**) ADEMA (Algorithm to Determine Expected Metabolite Level Alterations Using Mutual Information) analyses to compare metabolite concentrations in *H3.3-K28M*-mutated samples (relapse versus diagnostic), and BT69, BT68, and BT83 PDCLs (hypoxic versus normoxic environment). (**b**) Metabolites’ modulation in 6 relapses versus 10 diagnostic pHGGs bearing an *H3.3 K28M* driver mutation. (**c**) PET (positron emission tomography) scans at diagnosis and relapse in BT69, BT68, and BT83 patients. 2HG: 2-hydroxyglutarate; GABA: gamma-aminobutyric acid; GPC: glycerophosphocholine; GSH: gluthatione; PC: phosphocholine; FDG: fluorodesoxyglucose); ROS: reactive oxygen species.

**Figure 3 cancers-11-01875-f003:**
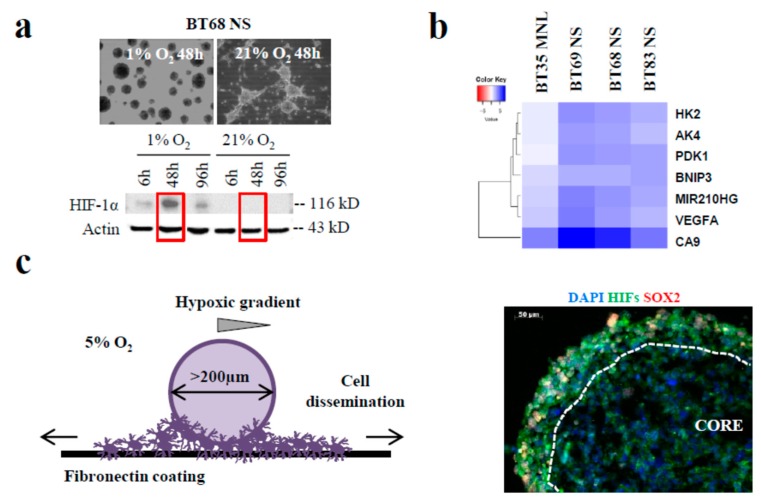
In vitro modeling of cell dissemination from hypoxic PDCL spheroids to recapitulate tumor progression in a patient. (**a**) Phase-contrast images and HIF-1α expression of BT68 NS in normoxic and hypoxic conditions at the indicated times. (**b**) Significant differential expression of seven genes involved in the hypoxia signaling pathway in cells exposed to 21% O_2_ versus 1% O_2_. (**c**) Diagram depicting a model of the cell dissemination from hypoxic spheroids and the immunofluorescent staining of HIFs and SOX2 (SRY-Box2) in the core and the periphery of the spheroid. (**d**) Quantification of the number of migrating cells. (**e**) Phase-contrast micrographs of PDCL migration following crystal violet staining. (**f**) Injected T1-weighted magnetic resonance images of patient tumors at relapse. Red arrows—distant metastasis. **** *p* < 0.0001; * *p* < 0.05. Scalebar—200 µm. DAPI: 4′,6-diamidino-2-phénylindole.

**Figure 4 cancers-11-01875-f004:**
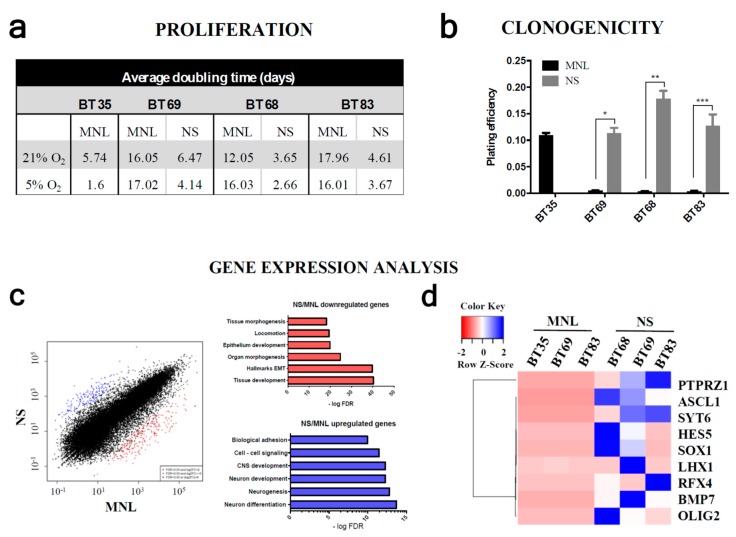
In vitro and in vivo growth advantage and drug sensitivity of a neurospheres culture compared to the paired monolayers. (**a**) Evaluation of the PDCL doubling time at 21% and 5% oxygen. (**b**) Quantification of colony counts in PDCLs. (**c**) Scatter plot showing upregulated genes in blue and downregulated genes in red, in NS compared to MNL cells (log2 fold change > or <6) and the corresponding functional annotation. (**d**) An unsupervised hierarchical clustering from all PDCLs samples according to a gene set implicated in stem-like cells potential was calculated using the UPGMA algorithm (unweighted pair group method with arithmetic mean) with the SERE (simple error ratio estimate) coefficient as the distance measure. (**e**) Hematoxylin & Eosine staining and survival curves of immunodeficient mice injected with BT68 MNL and BT68 NS cells. (**f**) Evaluation of the BT69 and BT83 responses to temozolomide (TMZ) using the IncuCyte^©^ system. Scalebar—100 µm. MNL: monolayer, NS: neurosphere. Data are the mean ± s.e.m. of three independent experiments. *** *p* < 0.0001; ** *p* < 0.001; * *p* < 0.05. PTPRZ1: receptor-type tyrosine-protein phosphatase zeta; SYT6: synaptotagmin 6; HES5: hes family bHLH transcription factor 5; SOX1: SRY-box transcription factor 1; LHX1: LIM homeobox 1; RFX4: regulatory factor X4; BMP7: bone morphogenetic protein 7; OLIG2: oligodendrocyte transcription factor 2.

**Figure 5 cancers-11-01875-f005:**
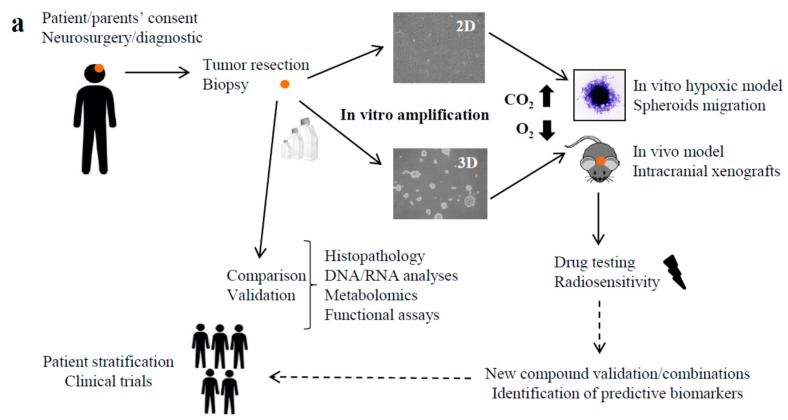
Standardized hypoxic PDCL approach. (**a**) Strategy for pre-clinical drug testing using patient-derived models. (**b**) Proposed hierarchical model of patient-derived cell expansion. (**c**) Co-cultures of 3D and paired 2D cell line. Phase contrast images of BT83 3D culture, 2D culture, and 3D/2D co-cultures expanded for 20 days (×100).

**Table 1 cancers-11-01875-t001:** Clinical characteristics of four pediatric *H3F3A-K28M-*mutated high-grade gliomas. BT: brain tumors; HGG: grade IV glioma; DIPG: diffuse intrinsic pontine glioma; Iri/Beva: Irinotecan/Bevacizumab; ADC: apparent diffusion coefficient measured within T2-weighted injected sequences. Ki67, HIF1 (hypoxia inducible factor 1), GFAP (glial fibrillary acidic protein), EGFR (epidermal growth factor receptor), PTEN, and Olig2 stainings (% of positive cells in tumor specimen) were used. Tumors for which analyses were unavailable are designated NA (not available).* Stupp protocol with a total of 12 courses of temozolomide was used in post-radiotherapy.

Patients	BT35	BT69	BT68	BT83
**Sample**	Relapse	Relapse	Diagnosis	Diagnosis
**Location**	Thalamic	Thalamic	Pons	Pons
**Age (years)**	18	14	9	13
**Diagnosis**	HGG	HGG	DIPG	DIPG
**1st line therapy**	Stupp protocol	Stupp protocol	RT + cilengitide	RT + dasatinib
**2nd line therapy**	RT + Iri/Beva	RAPIRI protocol	No treatment	RAPIRI protocol
**Survival (months)**	12	17	10	9
	**Imaging**			
**Contrast-enhanced T1-weighted MRI**	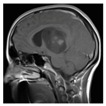	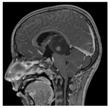	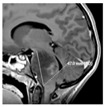	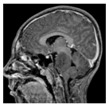
**ADC (mm(2)/s)**	1047	1029	945	1068
**T2-weighted FLAIR MRI**	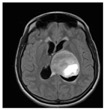	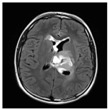	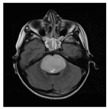	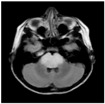
	**Histology (diagnosis)**			
**H&E**	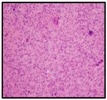	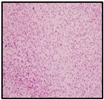	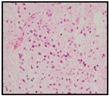	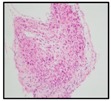
**IHC**				
Ki67 (%)	50–80	20–25	15	10–15
GFAP	pos	pos	pos	pos
p53	pos	neg	pos	pos
EGFR	neg	pos	pos	pos
PTEN	pos	pos	NA	pos
HIF-1α (%)	20	50	50	20
Olig2 (%)	70	100	10	50
Olig2 (%) at relapse	30	50	NA	NA

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
