# Peer review of "Hypoxic Environment and Paired Hierarchical 3D and 2D Models of Pediatric H3.3-Mutated Gliomas Recreate the Patient Tumor Complexity"

_cancers, 2019, doi:10.3390/cancers11121875_

Round 1

Reviewer 1 Report

This interesting work explores the impact on neurosphere culture condition versus monolayer culture conditions in the establishment of primary cell culturs fom pediatric high grade gliomas (pHGGs). The authors found that the cells retained key characteristics like tumorigenicity when implanted in NSG mice and sensitivity against TMZ when cultivated as neurospheres vs. monolayer culture under mildly hypoxic conditions (5% O2). These findings are highly interesting, because the issue of suitable in vitro models in neuro-oncology is still not resolved yet.

As I understood, the authors have used DMEM/F12 medium with 10% serum for the monolayer culture. But then the effects seen may likely not attributed to the monolayer culture conditions itself, but to the differentiation induced by the serum. Have you tested culture conditions with laminin coated plates and serum-free medium containing B27? This issue must be addressessed or discussed prior to publication.

Author Response

Dear Reviewer 1, 

First of all, thanks for all the comments and recommendations for our manuscript entitled: Hypoxic environment and paired hierarchical 3D and 2D models of pediatric H3.3 mutated gliomas recreate the patient tumor complexity.

All the modifications done in the manuscript are written in red color. I apologize as I had only my personal laptop during my current holidays and could not do track changes as required. I hope it will be suitable like this. All corrections were approved by authors.

For your question around media of the cultures, we tried several concentrations of serum with DMEM/F12 and we had the same characteristics and proliferation. We did a laminin coating for the 2D culture initiation. Usually, we stopped the laminin coating after few passages (usually around 5 to 6). For this purpose, I addressed this item in the discussion part as we will not show all the cultures done in main figures. We did laminin coating without serum, but it did not work for the culture.

I hope it will answer appropriately your question.

Sincerely

Reviewer 2 Report

The manuscript by Blandin et al. investigated multiple aspects of H3 K27M-mutant gliomas and could provide interesting data about these rare tumors, but presentation of the results must be significantly improved and other issues should also be addressed.

Introduction: I would move most of the last part of the Introduction to the Discussion since you extensively comment on the study findings. I would just leave an explanation of the study aims and an overall description of the experimental strategy.

Table 1 seems to be missing, so it is difficult to evaluate the characteristics of the included samples.

By reading the first paragraph of the results it seems that only 4/11 samples were investigated in the study, but by looking at Figure 2 (HRMAS NMR) there is a n=10 and then in Figure 2B legend there is a reference to 6 relapses. It should be clear which samples were used for each analysis and why. Also, the "flow" of the results should be clearer: why you moved from experiment A to B and so on? For example, the part regarding the genotype and phenotype of PDCL cultures seems a bit extraneous from the other results.

How you selected the 4 samples (out of 11) used to establish cell lines? This should be specified.

Which samples were analyzed at initial diagnosis and at relapse? After how long and which treatments patients did relapse and underwent the new surgery?

Results: you could comment a bit on the meaning/significance of the observed results. For example, in line 214-215 "Transcriptomic analyses identified 7 upregulated genes in PDCLs exposed to 1%O2 and no downregulated genes...": do you think these genes have a specific meaning? Do they give us new information regarding the tumor?

The Discussion starts dealing with temozolomide and the therapeutic relevance of the results, but these were the last part of the results. I would first comment on the other results and then about their potential relevance for treatment. Again, the whole "vision" of the study should be clearer through Introduction, Results and Discussion.

Figure 1C is too small: you could provide the main results in the figure and a larger version of the Circos plots in the supplementary material.

Please check that every abbreviation is defined at its first use.

Some minor language/spelling errors are present (e.g. line 16, "harborred" instead of "harbored", line 425, "RNAsequencing" is missing a space). Please check.

Author Response

Dear Reviewer 2,

First of all, thanks for all the comments and recommendations for our manuscript entitled: Hypoxic environment and paired hierarchical 3D and 2D models of pediatric H3.3 mutated gliomas recreate the patient tumor complexity.

All the modifications done in the manuscript are written in red color. I apologize as I had only my personal laptop during my current holidays and could not do track changes as required. I hope it will be suitable like this. All corrections were approved by authors.

As recommended, we changed the end of the introduction just specifying the techniques and the different steps of the study with our hypotheses.

We add in the manuscript directly the table 1 after the first paragraph of results’ part. We provided also the figures’ corrections in a separated ppt file to provide a higher pixelization of the figures. We grouped table 1, figures and supplementary figures in a specific file with this resubmission. At the first submission, we had separated files : table 1 and figures.

For the sample numbers, we precisely explained in the manuscript that in the PEDIAMODECAN program (PDCLs and PDX program), we had 7 DIPGs with 4 derived cell lines and only 2 were H3.3 K28M-mutated cell lines. We were also able to derived 2 out 4 tHGGs’ cell lines, where we had the same driver tumor mutation, H3.3 K28M. For the samples used in metabolomics, we had 6 tHGGs with the matched relapse samples. 2 out of those 6 tHGGs were included previously to the PEDIAMODECAN program. Only 4 out of the 7 DIPGs were assessed by HRMAS as they were H3.3 K28M mutated. The 3 others were H3.1 mutated and the remaining DIPG specimen was a wild type tumor. The 10 H3.3 K28M-mutated samples analyzed in the metabolomic part were comprising those where we were able to derived a cell line (4 PDCLs), those where we were not able to derived a cell line (4) and 2 where the surgery was done before the PEDIAMODECAN program.  We put a simple explanation in the results’ section with metabolomic results.

For the sample selection, we did select the 4 samples because of their driver mutation. It was specified in the clinical results’ paragraph. We had a 50% success for cell line derivation. All tHGGs’ relapses were collected after 1st recurrence. The table 1 is specifying the different treatments. All 6 tHGGs were treated based on the Stupp protocol with 12 courses of temozolomide post-radiotherapy or the standard arm of HERBY protocol.

We detailed the flow of the results adding sentences explaining and detailing the different steps of our study. We also detailed further several results as suggested by the second reviewer especially for the transcriptomic analyses in hypoxia.

We modified the discussion as required and changed the flow of this specific part. We added specific sentences to explain where we wish to go for 2D/3D standardization.

For the Figure 1C, we provided as suggested a larger version of the circus plots as a supplementary material (supplementary figure 1). We changed thereafter the numbers of the different supplementary figures in the figure ppt and in the manuscript (all those corrections are also in red in the word file).

We checked the abbreviations and the spelling errors to afford a well written manuscript. 

Hoping to have answered all requirements

Sincerely

Round 2

Reviewer 1 Report

With the extensive changes done to the manuscript, this is now a highly worthy contribution to the scientific literature. I strongly recommend publication.

Author Response

Dear Reviewer

We let our native english colleague in the Lab review once again our manuscript to finalize and change every spelling or typing error.

Thanks for your support

Sincerely 

Natacha Entz-Werlé and Anne Florence Blandin

Reviewer 2 Report

I thank Authors for their revision: I think they have sufficiently addressed the suggestions.

Some minor language issues could still be addressed to improve readability (e.g. line 131: " We were able to derive 4 out 7" should be "We were able to derive 4 out of 7", line 226: "PET-scaning" should be "PET-scanning", line 392: "another" should be "other", line 403: "choose" should be "chose", line 468: "Diagnosrics" should be "Diagnostics").

Author Response

Dear Reviewer

As required, we let our native english colleague read again our manuscript to check every spelling or typing errors and take into account your suggestions. We tried to improve as recommended the manuscript to clearly present our results. 

Thanks for your suggestions and for your support

Sincerely

Natacha Entz-Werlé and Anne Florence Blandin